# Ethylene-Vinyl Acetate (EVA) Containing Waste Hemp-Derived Biochar Fibers: Mechanical, Electrical, Thermal and Tribological Behavior

**DOI:** 10.3390/polym14194171

**Published:** 2022-10-04

**Authors:** Maria Giulia Faga, Donatella Duraccio, Mattia Di Maro, Riccardo Pedraza, Mattia Bartoli, Giovanna Gomez d’Ayala, Daniele Torsello, Gianluca Ghigo, Malucelli Giulio

**Affiliations:** 1Institute of Sciences and Technologies for Sustainable Energy and Mobility, National Council of Research, Strada delle Cacce 73, 10135 Torino, Italy; 2Department of Chemistry, University of Torino, Via Pietro Giuria, 7, 10125 Torino, Italy; 3Center for Sustainable Future Technologies, Italian Institute of Technology, Via Livorno 60, 10144 Torino, Italy; 4Consorzio Interuniversitario Nazionale per la Scienza e Tecnologia dei Materiali (INSTM), Via G. Giusti 9, 50121 Firenze, Italy; 5Institute for Polymers, Composites and Biomaterials, National Council of Research, Via Campi Flegrei 34, 80078 Pozzuoli, Italy; 6Politecnico di Torino, Department of Applied Science and Technology, Corso Duca degli Abruzzi 24, 10129 Torino, Italy; 7Istituto Nazionale di Fisica Nucleare, Sez. Torino, Via P. Giuria 1, 10125 Torino, Italy; 8Politecnico di Torino, Department of Applied Science and Technology, Viale Teresa Michel 5, 15121 Alessandria, Italy

**Keywords:** EVA, hemp fibers, biochar, circular bioeconomy, thermal properties, mechanical behavior, electrical conductivity, wear behavior

## Abstract

To reduce the use of carbon components sourced from fossil fuels, hemp fibers were pyrolyzed and utilized as filler to prepare EVA-based composites for automotive applications. The mechanical, tribological, electrical (DC and AC) and thermal properties of EVA/fiber biochar (HFB) composites containing different amounts of fibers (ranging from 5 to 40 wt.%) have been thoroughly studied. The morphological analysis highlighted an uneven dispersion of the filler within the polymer matrix, with poor interfacial adhesion. The presence of biochar fibers did not affect the thermal behavior of EVA (no significant changes of Tm, Tc and Tg were observed), notwithstanding a slight increase in the crystallinity degree, especially for EVA/HFB 90/10 and 80/20. Conversely, biochar fibers enhanced the thermo-oxidative stability of the composites, which increased with increasing the biochar content. EVA/HFB composites showed higher stiffness and lower ductility than neat EVA. In addition, high concentrations of fiber biochar allowed achieving higher thermal conductivity and microwave electrical conductivity. In particular, EVA/HFB 60/40 showed a thermal conductivity higher than that of neat EVA (respectively, 0.40 vs. 0.33 W·m^−1^ ·K^−1^); the same composite exhibited an up to twenty-fold increased microwave conductivity. Finally, the combination of stiffness, enhanced thermal conductivity and intrinsic lubricating features of the filler resulted in excellent wear resistance and friction reduction in comparison with unfilled EVA.

## 1. Introduction

The paradigm of a “zero waste” circular bioeconomy, popular in research, industrial and governance circles, has drawn ever-increasing interest for employing waste bio-masses to obtain valuable materials for bio-refineries [1,2,3]. Among biomasses, agricultural residues are widely investigated since they represent a particularly promising alternative to ensure and meet the increasing global energy and materials demand [4,5,6].

In this contest, biochar, a carbonaceous and renewable substance obtained by the thermochemical conversion of biomasses and/or post-consumer agricultural wastes in an oxygen-limited environment has increasingly been explored as filler for the development of polymer-based composites [7].

Biochar has generally been exploited at the pre-industrial scale as soil amendment [8], though it has found many other applications ranging from environmental remediation [9], energy storage [10] and catalyst support [11]. Many research studies have highlighted that the addition of biochar particles results in polymer composites with improved flexural and tensile properties, enhanced impact strength and higher heat deflection temperature [7,12,13,14,15]. Additionally, it was demonstrated that biochar can bestow electrical conductivity, flame retardance and wear resistance both to thermoset and thermoplastic-based systems [16,17,18,19,20], demonstrating its potential to replace conventional and expensive carbonaceous fillers derived from fossil fuels.

Among thermoplastic polymers, ethylene-vinyl acetate (EVA) copolymer has become one of the most useful materials for electrical cables in the automotive industry [21,22,23]. Moreover, it has found applications for the production of sensors and self-regulating heaters [24,25], for electromagnetic interference shielding (EMI) [26], for the manufacturing of soles in the shoe industry and as an industrial coating, among others [21,26]. However, there are certain limitations to the use of neat EVA copolymer without additives/fillers because of its low thermal and electrical conductivity, easy flammability and occasionally insufficient mechanical and tribological properties.

The properties of EVA have been improved by using a wide variety of fillers. For electrical conductivity, for example, carbonaceous fillers have been incorporated into EVA with successful results [27,28,29,30,31]. Yuan et al. employed a polyaniline (PANI)/reduced graphene oxide (RGO) composite, prepared by in situ polymerization of aniline monomer in the presence of RGO [30]. They found that at 4.0 and 8.0 wt.% of RGO and PANI, respectively, EVA composites showed a dramatically enhanced conductivity (i.e., 1.07 × 10^−1^ S·cm^−1^), demonstrating their potential applications for electromagnetic (EMI) shielding, antistatic and corrosion-resistant coatings. Das et al. [31] studied EMI shielding efficiency of different EVA composites filled with commercial conductive carbon black (CB, loaded up to 50 wt.%) and short carbon fibers (SCF, from 5 to 30 wt.%). They concluded that EVA/SCF composites are more effective in providing EMI shielding than the counterparts filled with conductive carbon black.

EVA thermal conductivity can be improved by zinc oxide (ZnO) nanoparticles [32], boron nitride (BN) [33,34] and even short carbon fibers (SFC), carbon black (CB) and graphene (G) [28,35]. For example, Wang et al. used short carbon fibers for preparing EVA composites by means of a spatial confining forced network assembly method. They found that EVA/SCF composite samples presented a thermal conductivity as high as 19.68 W·m^−1^ ·K^−1^ under the optimal processing parameters [35]. Azizi et al. evidenced that the thermal conductivity of EVA/CB (7 wt.% of filler) and EVA/G (15 wt.% of filler) increased by 16 and 22%, respectively, compared to the unfilled copolymer [28].

There is also a fair number of publications dealing with the improvement of mechanical properties in EVA composites. Zec et al. [36] used ultra-high molecular weight polyethylene (UHMWPE, PE)/oxidized PE (ox-PE) fibers and alumina as reinforcement for the preparation of poly (ethylene-co-vinyl acetate) (EVA)/hydrolyzed EVA (EVAOH) composites. Basalt fibers were also successfully employed by Yao et al. after surface modification and blending with magnesium hydroxide [37]. Other fillers used for the same purpose were wood flour particles [38], cellulose microfibers [39] and natural fibers [40].

More limited is the study of the tribological behavior of EVA composites. Ravi Kumar et al. investigated the abrasive wear behavior of neat EVA and nanoclay-filled EVA/LDPE composites in dry conditions [41,42], demonstrating that neat EVA exhibited the best wear resistance. Inclusion of low-density polyethylene (LDPE) in EVA increased the specific wear rate. The addition of nanoclay and compatibilizer in an EVA/LDPE blend significantly improved the abrasion resistance of EVA/LDPE.

In this light, the aim of this study is to investigate composites based on EVA copolymer and biochar obtained from the pyrolysis of waste hemp short fibers. This biochar (coded as HFB—hemp fiber biochar) has been used for the first time as filler in this copolymer. The use of waste HFB, a renewable substance, would contribute to the reduction of carbon components sourced from fossil fuels for the preparation of composites and to the reduction of waste textile fibers that represents a big challenge [43]. The mechanical, tribological, electrical (DC and AC) and thermal properties of EVA/HFB composites containing different amounts of fibers (ranging from 5 to 40 wt.%) have been thoroughly studied. The potential application of these composites is in the automotive sector as injection-molded components/parts with tunable properties.

## 2. Materials and Methods

### 2.1. Materials

The polymer matrix, an ethylene-vinyl acetate copolymer (EVA Greenflex MQ40), with melt flow index (MFI) of 12 g/10 min at 190 °C/2.16 kg (ISO 1133), containing 19% of vinyl acetate monomer (VA), was purchased by Versalis S.p.A (Mantova, Italy). The biochar used as filler (HFB) was obtained from the pyrolysis of waste short damaged hemp fibers. These hemp fibers, kindly provided by Assocanapa [44], were less than 10 cm long and were not suitable for other applications. They were collected and used without further treatments.

### 2.2. Methods

#### 2.2.1. Waste Hemp Fiber Pyrolysis and Preparation of Composites

The pyrolysis of the waste hemp fibers was carried out at 1000 °C in a tubular furnace (Carbolite TZF 12/65/550) under N_2_ atmosphere (flow rate: 0.4 mL/min), with a heating rate of 15 °C/min. The temperature was kept at 1000 °C for 30 min; then, it was decreased to room temperature in N_2_ atmosphere.

EVA/HFB composites were prepared by melt-mixing, using a Brabender W50E apparatus (Plasti-Corder, Duisburg, Germany). The raw materials were poured inside the melting chamber, previously heated at 140 °C and mixed at 70 rpm for 5 min. Composites with different HFB weight loadings (namely, 5, 10, 20, 30 and 40 wt.%) were prepared. Neat EVA was processed in the same conditions for comparison. The obtained materials were chopped for reducing their size and then compression-molded by using a Collin P200T laboratory press (Maitenbeth, Germany) at 140 °C for the first 4 min without applied pressure and for another 4 min with the pressure raised to 100 bar. Samples, 1 mm thick, were used for the DSC, TGA and tensile tests. For conductivity and tribological tests, specimens with a rectangular shape and dimension of 31 × 31 × 3 mm^3^ and 20 × 20 × 4 mm^3^, respectively, were prepared.

#### 2.2.2. Scanning Electron Microscopy

Scanning electron microscopy (SEM, ZEISS EVO 50 XVP—Oberkochen, Germany—with LaB_6_ source) was exploited for investigating the morphology and the distribution of the HFB into the polymer matrix. The cross section of the samples was examined after cryogenic fracture in liquid nitrogen. SEM was also employed for observing the worn surfaces after the tribological tests. Before the observation, samples were gold-metalized with a layer of about 10 nm to avoid any charge effect.

#### 2.2.3. Thermal Analysis: Differential Scanning Calorimetry (DSC) and Thermogravimetric Analysis (TGA)

The thermal properties of the compression-molded composites were investigated with a differential scanning calorimeter (DSC, Q2000, TA Instruments, New Castle, DE, USA, Mettler DSC822). Calibrations, for both the temperature and the enthalpy, were performed by using pure indium as standard (T_m_ = 156.4 °C; ΔH_m_ = 28.15 J/g). About 8 mg of the sample were analyzed to measure the melting temperature (T_m_), the glass transition temperature (T_g_), the crystallization temperature (T_c_) and the degree of crystallinity (%) of the investigated systems. The measurements were carried out in a nitrogen atmosphere (30 mL/min) with a heating and cooling rate of 10 °C/min according to the following cycle: from −80 °C to 120 °C (first heating run), from 120 °C to −80 °C (cooling run) and again from −80 °C to 120 °C (second heating run). The crystallinity degree of EVA and its composites was calculated as:χ_c_ = (ΔH_m_/ΔH^0^_PE_) × 100
where ΔH_m_ is the measured enthalpy of melting, and ΔH^0^_PE_ is the enthalpy of melting per gram of 100% crystalline PE (ΔH^0^_PE_ = 277.1 J·g^−1^), as crystallinity is referred to polyethylene segments only. The glass transition temperature was determined as the maximum of the peak obtained by applying the first derivative procedure.

Thermogravimetric analyses were performed by using a TGA Discovery (TA Instruments, New Castle, DE, USA). Samples of about 11 mg were heated from 30 °C to 700 °C with a heating rate of 10 °C/min. T_10%_ (10% weight loss), and T_max_ (maximum rate of degradation measured as the peak of the first TG derivative) were evaluated. Measurements were carried out both in N_2_ and air atmosphere (flow rate: 45 mL/min).

#### 2.2.4. Thermal Conductivity

The thermal conductivity was evaluated by using a TPS 2500S apparatus ((Hot Disk AB, Göteborg, Sweden) with a Kapton sensor (radius 3.189 mm). Two specimens (30.6 × 30.6 × 3 mm^3^) were placed on opposite sides of a probe, able to emit power pulses and record the time that the material takes to return to the initial temperature. A pulse, with a power of 50 mW, was used to analyze these materials with 2 s acquisition time. Five different acquisitions were performed for each specimen with an interval of 10 min between the acquisitions to allow the specimen to restore the starting conditions. The measurements were carried out in a water bath (Haake A40, Thermo Scientific Inc., Woburn, MA, USA) with a thermostat set at 23 ± 0.01 °C equipped with a temperature controller (Haake AC200, Thermo Scientific Inc., Woburn, MA, USA).

#### 2.2.5. Electrical Conductivity

The electrical conductivity was measured under increasing pressure loads from 1 bar up to 1500 bar, using a Specac Atlas Manual Hydraulic Press 15T (Orpington, UK), according to Giorcelli et al. [18]. To ensure the material conductivity, electrically insulating sheets were placed between the conductive cylinders and the loading surfaces. The resistance of the composites was assessed using an Agilent 34401A multimeter (Keysight Technologies, CA, USA). As indicated by the method reported by Torsello et al. [45], the complex permittivity of the samples was measured in the GHz range by means of an EpsiMu toolkit [46], a cylindrical coaxial cell containing the sample as a dielectric spacer between inner and outer conductors, whose diameters were 0.3 cm and 0.7 cm, respectively. To avoid mismatch and energy loss, the cell was connected to standard connectors, which allowed maintaining the characteristic impedance at 50 Ω. The measurements were analyzed with a two-port transmission line technique, using a Rohde Schwarz ZVK Vector Network Analyzer, properly calibrated. A Nicolson–Ross–Weir transmission/reflection algorithm was exploited for determining the electromagnetic properties of the materials [47].

#### 2.2.6. Tensile Tests

Mechanical tests were performed with an Instron 5966 dynamometer (Norwood, MA, USA) in tensile configuration. Specimens were 62 × 3.15 × 1 mm^3^. The clamps were positioned at 3 cm, defining the elongation zone. The test was performed at room temperature. The method applied was: 1 mm/min of speed until 0.2% of deformation, then 50 mm/min until breaking. Young’s modulus (*E*), elongation at break (%) and tensile strength (MPa) were recorded. The mechanical parameters were averaged over ten independent measurements for each type of composite. One-way ANOVA tests and a Spearman correlation test with a significance level of 0.05 (*p* < 0.05) were carried out using Excel™ software (Microsoft Corp., Redmond, WA, USA) and the “Data analysis” tool.

#### 2.2.7. Tribological Tests

Tribological tests were carried out with a CSEM pin-on-disk Tribometer (Alpnach, Switzerland) in ball-on-disc configuration to measure the coefficient of friction (µ) and the wear behavior of the materials. The static component was a 6 mm ball of alumina (Al_2_O_3_). The tests were carried out under a normal load of 2 N and a linear speed of 10 cm/s (rotational radius of 5 mm). The sliding distance was fixed at 1099 m (number of laps 35,000), and tests were performed at room temperature. After the measurements, wear depth and volume loss were measured by a contact profilometer (Form Talysurf 120, Taylor Hobson, Leicester, UK). From the obtained 3D profiles, 2D profiles were extracted from different locations, and the wear volume was calculated by integrating the surface area of the tracks over circumference length. The specific wear rate “*K*” (mm^3^/m·N) was calculated by applying the following formula [48]:$$K=\frac{V}{S\cdot L}$$
where “*V*” is the wear volume (mm^3^), “*S*” is the total sliding distance (m), and “*L*” is the load (N). Worn surfaces were analyzed by SEM-EDS.

## 3. Results and Discussion

### 3.1. Morphological Analysis

The morphology of the pyrolyzed hemp fibers was already reported in a previous work [20]. Briefly, pyrolyzed fibers were several millimeters long and had a diameter of about 50 μm. Regarding the morphology of the composites, all the composites showed a bad distribution of filler (indicated by red arrows) in the matrix together with a poor polymer/filler interaction, also evidenced by the presence of voids, many of which were generated by the debonding of the fibers from the polymer matrix because of the material fracture (Figure 1A,B). Some characteristic features of EVA/HFB 95/5 and 60/40 fracture surfaces are shown in Figure 1C,D, respectively. A biochar fiber, whose pores are filled by the polymer, is well depicted in Figure 1C. However, not all the pores are filled with EVA chains. At high concentration (Figure 1D), biochar fibers were more aggregated and appear oriented in the direction of the fracture surface (some of them are indicated by the red arrows).

### 3.2. Thermal Analyses

DSC thermograms of EVA and its composites are shown in Figure 2; Table 1 collects the thermal parameters. In the first heating run, neat polymer (Figure 2A, black curve) showed two distinct melting peaks, at 50.1 °C (T_m1,1_) and 84.5 °C (T_m1,2_), respectively. This behavior, well described in the literature [49,50,51], is due to the presence of the vinyl acetate (VA) fraction, which is not able to crystallize, and that leads to the formation of polyethylene chains with a different ability to crystallize, melting at different temperature. In the crystallization cooling run (Figure 2B, black curve), when EVA crystallized from the melt, the long polyethylene chain segments with the strongest crystallization ability started to crystallize first, hence originating the main crystallization region (primary crystallization, peak at 68.1 °C) [49,50,51]. The non-crystalline VA groups were gradually excluded from the crystallization zone, while the short polyethylene chain segments with weak crystallization ability formed imperfect crystals at the interface between the crystal and the amorphous zones (secondary crystallization, peak at 39.7 °C) [51,52]. However, the crystallization ability of the short polyethylene chain segment, in the DSC experiment conditions, was very limited; therefore, the secondary crystallization was barely visible in the cooling trace. Following this behavior, the second heating run (Figure 2C, black curve) showed a main melting peak at a higher temperature (T_m2_ ≈ 85.4 °C), whereas the lower melting peak appeared as a shoulder at about 66 °C. As far as the thermal behavior is concerned, the composites did not evidence substantial differences with respect to the unfilled copolymer, indicating that biochar hemp fibers do not influence the melting/crystallization process of EVA. The glass transition temperature (T_g_) of neat EVA, evaluated by the first derivative of the first heating curve, was −24.0 °C [50]. Again, no significant variation of the T_g_ due to the presence of biochar fibers was detected for the composites, as reported in Table 1 and shown in the inset of Figure 2A. These findings can be ascribed to the poor interaction between the filler and the polymer matrix.

The crystallinity degree (χ_c1_) of the materials was evaluated from the ΔH_m1_ values measured in the first heating run. Neat EVA showed a crystallinity degree of 21.8%, in agreement with that found in the literature for copolymers containing the same amount of VA monomer (i.e., 19% of VA) [46,47]. The presence of biochar fibers did not significantly influence the crystallinity degree of the polymer matrix: indeed, the crystallinity increase observed for the composites containing 10–40 wt.% of HFB was very limited.

The thermogravimetric curves for EVA and its composites in N_2_ and air are shown in Figure 3A,B, respectively. The collected data are presented in Table 2. In nitrogen (Figure 3A), EVA copolymer showed a two-step degradation behavior. The first step (with T_max1_ = 356 °C), corresponded to the loss of the acetoxy groups coming from vinyl acetate units. The wt.% loss after the first degradation step was a qualitative index of the amount of VA in the copolymer. In this work, the obtained value was about 15 wt.%, which was in agreement with the amount of VA indicated in the data sheet of the supplier. The second degradation step (with T_max2_ = 477 °C) refers to the break of the polyunsaturated chains generated after the first degradation step [53]. As far as the composites are considered, no remarkable influence of the biochar fibers was observed in the two degradation steps and in T_max1_ and T_max2_ values (Table 2). Only the composite EVA/HFB 60/40 showed a T_10%_ value about 20 °C higher with respect to that of neat EVA and the other composites. Lastly, the residue at 700 °C was in good agreement with the HFB amount used for the preparation of the composites.

Under oxidative conditions (Figure 3B), EVA showed a degradation behavior quite similar to that observed in inert atmosphere [54]. Two degradation steps can be seen, with T_max1_ and T_max2_ at 342 °C and 440 °C, respectively. However, the presence of biochar fibers strongly influenced the degradation of the polymer matrix even at low filler loadings as already observed for other termoplastic polymer/biochar composites [55]. In particular, T_max1_ and T_max2_ increased with increasing the HFB loading, reaching the values of 353 °C and 480 °C, respectively, for the EVA/HFB 60/40 composite. Moreover, a rapid decomposition in a narrow temperature range was observed for neat EVA copolymer, whereas it became broader when the biochar loading increased. The degradation in air of the polymer chains led to the formation of hydroxyl and alkoxyl radical species, which in turn favored the scission of the C–C bonds of the polymer matrix and therefore the complete degradation of the material [56,57].

This oxidative degradation also involved HFB, which, through its C–C bonds, degraded with the same mechanism of the polymer matrix [57]. The ability of HFB to consume part of the free radicals produced by the thermo-oxidative process slowed down the degradation of the polymer matrix, which occurred at increasingly higher temperatures and with broader dTG curves when the HFB loading increased. Finally, the residue values obtained at 700 °C rose with the filler content, achieving 3 wt.% for the EVA/HFB 60/40 composite.

### 3.3. Thermal Conductivity Measurements

The thermal conductivity values as a function of the HFB loading are presented in Figure 4. Neat EVA showed a thermal conductivity of 0.33 W·m^−1^ ·K^−1^, which is comparable with the results reported in the literature for a copolymer containing 18% of VA [51]. The thermal conductivity slightly decreased with increasing the HFB loading and reached a minimum value of 0.27 W·m^−1^ ·K^−1^ when the amount of HFB was 10 wt.%. By further increasing the filler loading, the thermal conductivity increased, becoming even higher (0.40 W·m^−1^ ·K^−1^ for EVA/HFB 60/40) than that of neat EVA. At low filler loadings, the poor interfacial adhesion between polymer and filler (as observed by SEM images) and the presence of pores filled with air (air thermal conductivity is about 0.026 W·m^−1^ ·K^−1^) strongly reduced the effective thermal conductivity of the biochar fibers, causing a decrease in the thermal conductivity of the composites. However, when the amount of HFB increased, even if the interface and air pockets also increased, a higher amount of fibers was connected together and was oriented along the through-plane direction (as observed in Figure 1B), hence creating a more efficient heat path and determining an increase of thermal conductivity both for EVA/HFB 70/30 and 60/40.

### 3.4. Electrical Conductivity Measurements

All the composites showed insulating behavior for filler loadings below 40 wt.% (Figure 5). The EVA composites containing 40 wt.% of HFB showed conductivities ranging from 10^−7^ up to 10^−5^ S/m, which are relatively lower than those reported for other biochar composites, such as epoxy/hemp-derived biochar fibers [20], epoxy/coffee-derived biochar [18] and poly(propylene)/tea-derived biochar (produced at 1000 °C) composites [58], as summarized in Table 3.

The low conductivity values under the DC regime can be ascribed to a poor dispersion of the filler inside the matrix, as evidenced by SEM. This hypothesis is also supported by the high frequency electrical response of the composite (reported in Figure 6) both in the real part of the complex permittivity (ε’) and the conductivity (σ) as a function of frequency in the 100 MHz∓16 GHz range.

The real permittivity curves showed a slight initial decrease when frequency increased, followed by a plateau, whereas conductivity increased monotonically with frequency as previously observed for other polymer/HFB composites [45,59]. Furthermore, both ε’ and σ showed a quite linear increase with increasing HFB loading. Filler dispersion affected the frequency measurements carried out under DC regimes in a more remarkable way; the obtained results were in good agreement with the observations previously reported for other biochar-based composites. This strongly supports the poor dispersion of biochar fibers into EVA with an increment of percolation threshold.

### 3.5. Mechanical Properties

Young’s modulus (E), elongation and tensile strength at break of EVA and its composites are depicted in Figure 7A–C, respectively. As expected, the HFB presence strongly influenced the mechanical behavior of the composites. In particular, Young’s modulus (Figure 7A) increased significantly as the filler content increased, shifting from 56.3 MPa for neat EVA to 287.4 MPa for EVA/HFB 60/40, i.e., more than five times higher than that of unfilled EVA. This finding can be ascribed to the mechanical interlocking, created by the presence of an irregular-shaped filler mixed with the polymer matrix. Moreover, the addition of increasing amounts of a stiff filler accounted for an overall stiffening of the composites. The reinforcing effect of biochar on Young’s modulus of polymer-based composites is widely reported in the literature [14,60]. Moreover, the greater the amount of added HFB, the lower the elongation ability (Figure 7B) of the material because of the poor interface interaction between polymer and filler. Conversely, the tensile strength at break (Figure 7C) decreased with increasing the HFB amount. In particular, neat EVA exhibited a tensile strength of 17.6 MPa that decreased to about 4 MPa for the composites with 30 and 40 wt.% of HFB. This behavior, also reported in the literature [61], can be attributed to the higher number of HFB aggregates and inner defects acting as stress concentration points where cracks begin to develop. One-way ANOVA test with confidence level of 5% (*p* < 0.05) confirmed that the filler loading affected the mechanical properties. Spearman correlation coefficients showed a positive correlation between filler loading and Young’s modulus (0.897), while a negative correlation was confirmed for both elongation and strength at break with Spearman correlation coefficient values up to −0.980 and −0.925, respectively.

### 3.6. Tribological Tests

Figure 8 shows the coefficient of friction (COF) for EVA and its composites as a function of the sliding distance. The incorporation of pyrolyzed hemp fibers into the polymer significantly affected the COF. Neat EVA showed a COF of 1.18, which proportionally decreased with increasing the HFB content. EVA/HFB 60/40 displayed a COF equal to 0.61, i.e., 48% lower than that of unfilled EVA. The decrease in friction observed for the EVA composites was also accompanied by a significant reduction in the wear rate (Table 4) with respect to the unfilled matrix. These results can be ascribed to the nature of HFB, which, being a carbonaceous material, possesses self-lubricating properties, stabilizing the coefficient of friction and also reducing wear loss [62,63]. Moreover, the high thermal conductivity of carbonaceous components can accelerate the dissipation of the friction heat [64,65]. Our measurements, in fact, show higher values of thermal conductivity for the composites containing high amounts of biochar fibers (i.e., 30 and 40 wt.%). Further, as evidenced by mechanical results, the filler increased the stiffness of the composites, which, in turn, increased their wear resistance. Some typical SEM micrographs of the worn surfaces of neat EVA and EVA/HFB 60/40 and of the surfaces of their alumina counterbodies are shown in Figure 9: they confirm the observations previously discussed. In particular, the neat polymer (Figure 9A) showed severe wear losses characterized by a high degree of material deformation, whereas the worn surface of EVA/HFB composite (Figure 9B) was smoother and the signs of deformation were reduced. Some cracks appeared in the proximity of the biochar fibers due to the higher stiffness and lower ductility of the polymer composite with respect to the neat counterpart. By observing SEM micrographs of alumina balls against unfilled EVA (Figure 9C) and EVA composites (Figure 9D), transfer films appeared on both surfaces, but in a different amount: the higher amount of carbonaceous components in the adhesive film of the composites could be responsible for the decrease in the COF values.

## 4. Conclusions

Hemp fiber biochar-reinforced EVA composites were successfully prepared by melt-compounding. An uneven dispersion of the filler within the polymeric matrix was highlighted, with a low filler/matrix adhesion (i.e., characterized by the presence of voids generated by the detachment of the fibers following fracture propagation).

The presence of biochar fibers did not affect the thermal behavior of EVA (no significant changes of T_m_, T_c_, and T_g_ were observed), while it promoted a slight increase in the crystallinity degree, especially for EVA/HFB 90/10 and 80/20 systems. Conversely, HFB strongly affected the thermo-oxidative stability of EVA, which increased with increasing the HFB amount.

EVA/HFB composites showed higher stiffness and lower ductility than neat EVA. In addition, high concentrations of HFB allowed for achieving higher thermal conductivity and microwave electrical conductivity. In particular, EVA/HFB 60/40 showed a thermal conductivity higher than that of neat EVA (0.40 W·m^−1^ ·K^−1^ for EVA/HFB 60/40 and 0.33 Wm^−1^ K^−1^ for neat EVA), and EVA/HFB 40 showed an up to a twenty-fold increase in microwave conductivity. In particular, EVA/HFB 60/40 showed a thermal conductivity higher than that of neat EVA (0.40 W·m^−1^ ·K^−1^ vs. 0.33 W·m^−1^ ·K^−1^ respectively).

Finally, in the highly filled composites (i.e., incorporating 30 or 40 wt.% of filler), the combination of the self-lubricating properties of the filler with its high stiffness and thermal conductivity determined excellent wear properties, both in terms of COF and wear rate reduction when compared with the tribological features of neat EVA.

## Figures and Tables

**Figure 1 polymers-14-04171-f001:**
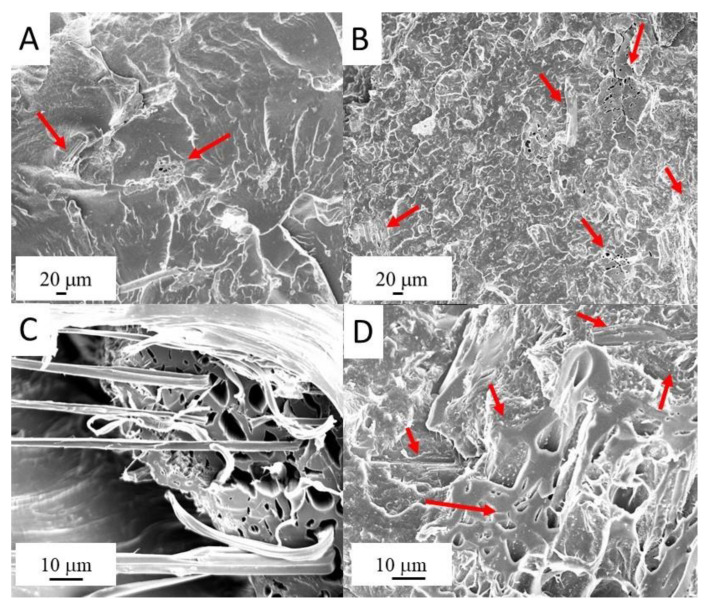
SEM micrographs of EVA/HFB 95/5 (**A**,**C**) and EVA/HFB 60/40 (**B**,**D**) cryogenically fracture surfaces.

**Figure 2 polymers-14-04171-f002:**
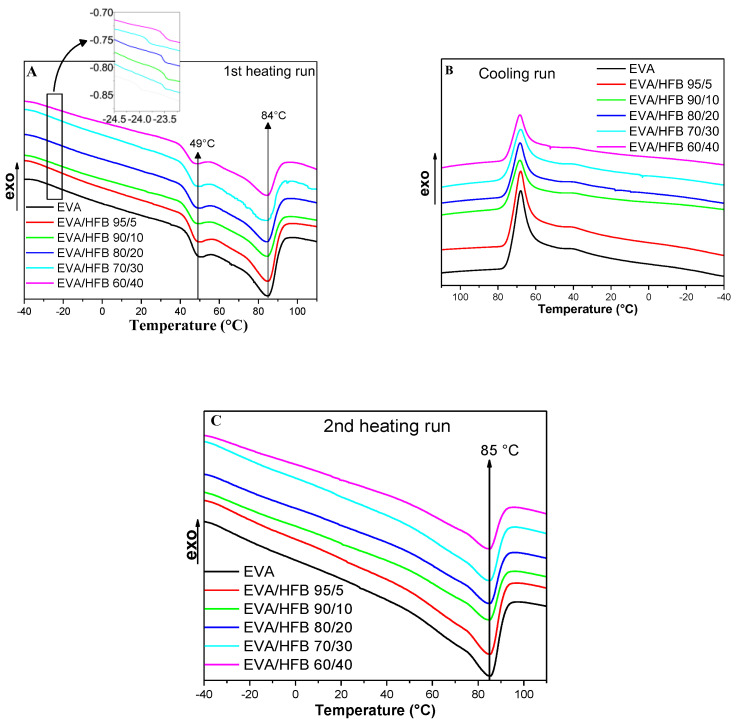
DSC curves of EVA and its composites: (**A**) first run from −40 °C to 120 °C; (**B**) cooling run from 120 °C to −40 °C; and (**C**) second heating run from −40 °C to 120 °C (heat flow rate: 10 °C/min).

**Figure 3 polymers-14-04171-f003:**
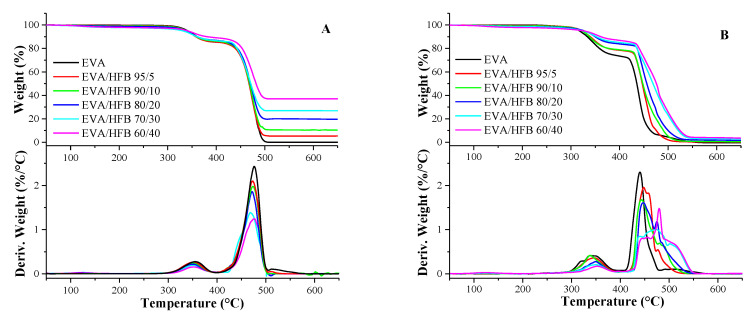
TG and dTG curves of EVA and its composites performed in: (**A**) nitrogen: and (**B**) air atmosphere (heating rate: 10 °C/min).

**Figure 4 polymers-14-04171-f004:**
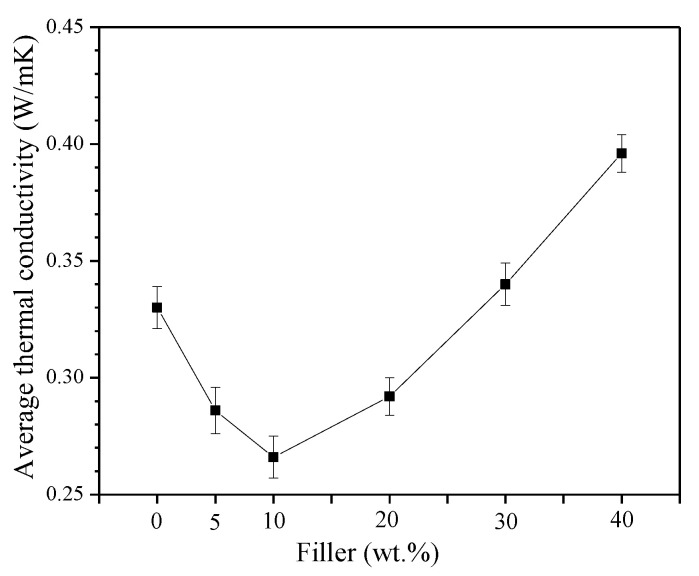
Thermal conductivity of EVA composites as a function of filler amount.

**Figure 5 polymers-14-04171-f005:**
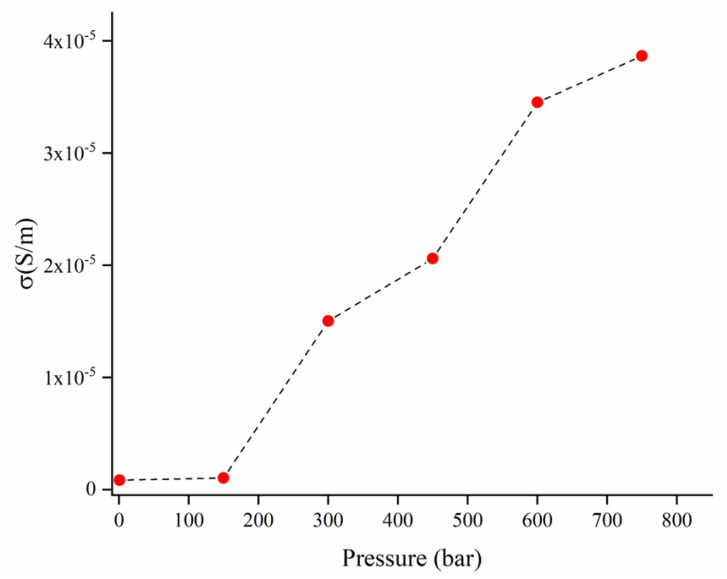
Conductivity of EVA/HFB 60/40 in the pressure range from 1 to 750 bar.

**Figure 6 polymers-14-04171-f006:**
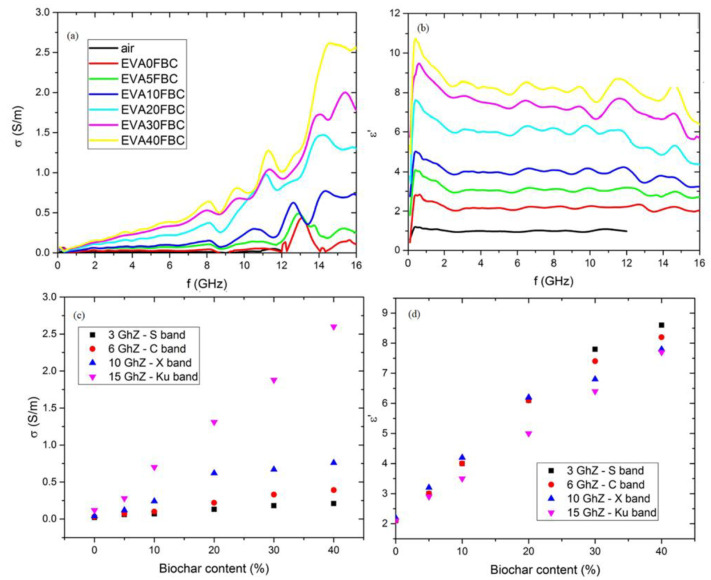
Conductivity (**a**) and real part of the complex permittivity (**b**) as a function of frequency; conductivity (**c**) and complex permittivity (**d**) as a function of HFB weight fraction at representative frequencies for communication microwave bands.

**Figure 7 polymers-14-04171-f007:**
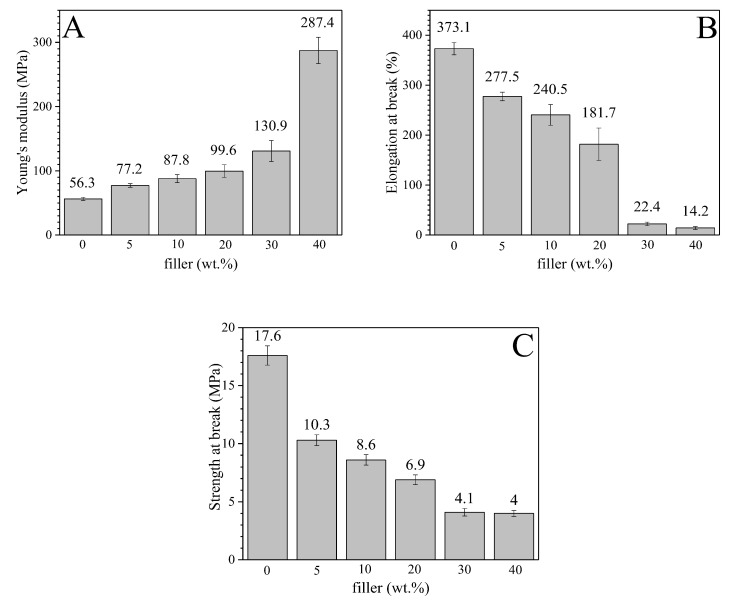
Tensile mechanical behavior of EVA/HFB composites: (**A**) Young’s modulus (MPa); (**B**) elongation at break (%); (**C**) tensile strength (MPa).

**Figure 8 polymers-14-04171-f008:**
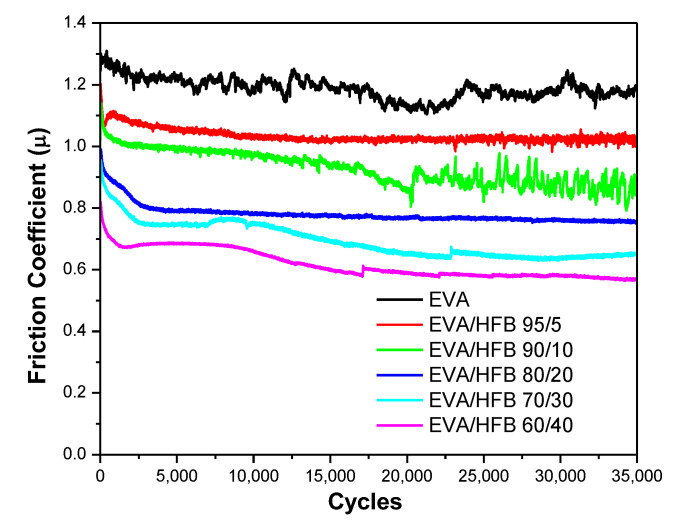
Friction coefficient of EVA and its composites.

**Figure 9 polymers-14-04171-f009:**
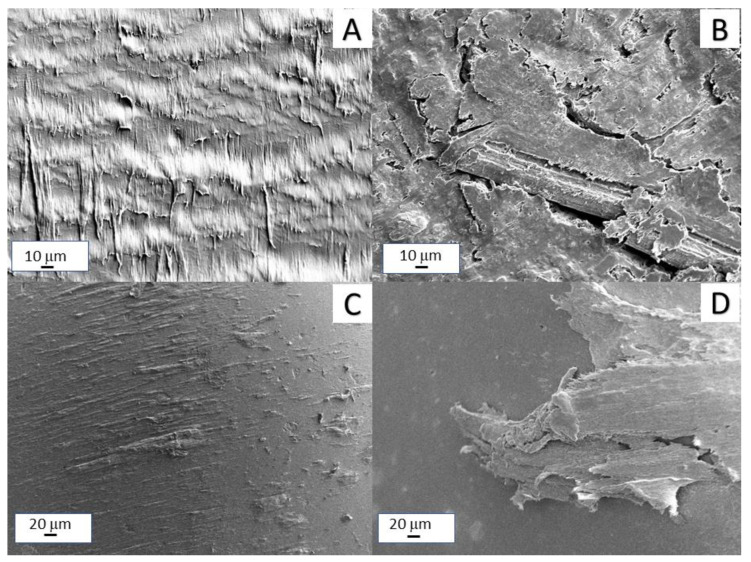
Topographical features of the surfaces of neat EVA (**A**); EVA/HFB 80/20 (**B**); alumina ball after sliding tests against neat EVA (**C**); alumina ball after sliding tests against EVA/HFB 60/40 composite (**D**).

**Table 1 polymers-14-04171-t001:** Thermal parameters of EVA and EVA composites.

Samples	T_g_ (°C)	T_m1,1_ (°C)	T_m1,2_ (°C)	T_c1_ (°C)	T_c2_ (°C)	T_m2_ (°C)	ΔH_m1_ (J/g)	χ_c1_ (%)
EVA	−24.0	50.1	84.5	68.1	39.7	85.4	60.5	22
EVA/HFB 95/5	−23.5	49.9	84.5	67.9	39.7	85.3	61.3	22
EVA/HFB 90/10	−23.5	48.8	83.9	68.3	39.9	85.0	66.5	24
EVA/HFB 80/20	−23.6	48.8	83.7	68.4	39.7	85.2	65.8	24
EVA/HFB 70/30	−23.8	48.4	83.3	68.2	39.6	84.9	64.1	23
EVA/HFB 60/40	−23.4	48.3	83.9	68.5	39.5	84.9	62.3	22

**Table 2 polymers-14-04171-t002:** Thermogravimetric data for EVA and its composites.

	N_2_				Air			
	T_10%_ (°C)	T_max1_ (°C)	T_max2_ (°C)	Residue @ 700 °C (%)	T_10%_ (°C)	T_max1_ (°C)	T_max2_ (°)	Residue @ 700 °C (%)
EVA	361	356	477	0.5	334	342	440	0
EVA/HFB 95/5	358	353	477	5.3	340	345	448	0.3
EVA/HFB 90/10	359	351	475	10.0	338	339	443	0.7
EVA/HFB 80/20	361	350	474	19.3	350	349	448	1.5
EVA/HFB 70/30	361	351	468	26.9	352	350	473	2.4
EVA/HFB 60/40	381	353	479	37.0	361	353	480	3.0

**Table 3 polymers-14-04171-t003:** Comparison of maximum conductivity values achieved by using different kinds of biochar.

Biomass Source	Matrix	Filler Loading (wt.%)	Conductivity (S/m)	Pressure (bar)	Ref.
Hemp	Epoxy resin	10	6	750	[20]
Hemp	EVA	40	10^−5^	750	This work
Coffee	Epoxy resin	15	10^−2^	1500	[18]
Tea	PP	40	2 × 10^−2^	1500	[58]

**Table 4 polymers-14-04171-t004:** Coefficients of friction (COF) and specific wear rates for EVA and its composites in a dry condition.

Sample	COF	Specific Wear Rate (mm^3^/N·m)·10^−4^
EVA	1.18 ± 0.03	10.6
EVA/HFB 95/5	1.03 ± 0.01	4.86
EVA/HFB 90/10	0.92 ± 0.05	2.95
EVA/HFB 80/20	0.77 ± 0.01	1.27
EVA/HFB 70/30	0.68 ± 0.05	0.79
EVA/HFB 60/40	0.61 ± 0.04	0.70

## Data Availability

Not applicable.

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
