# Peer review of "Ethylene-Vinyl Acetate (EVA) Containing Waste Hemp-Derived Biochar Fibers: Mechanical, Electrical, Thermal and Tribological Behavior"

_polymers, 2022, doi:10.3390/polym14194171_

Round 1
Reviewer 1 Report
This paper applied biochar obtained through pyrolysis from waste hemp fibers as fillers to prepare EVA-based composites with enhanced tribological properties, which interests readers from related field. It can be published after minor revision. Specific comments:
(1) "Biochar" is a short word and it is not nessesary to use an abbreviation, i.e. "BC", which would add ambiguities.
(2) The second paragraph of the Introduction is too long, which should be separated into several paragraphs.
(3) The authors discussed different characterizations of the EVA/Biochar composite, but what are the potential applications of this composite? If other fillers described in the Introduction section can already significantly improve EVA's properties, what are the reasons to use the fillers (biochar from hemp fibers) in this study to improve the different properties of EVA?
(4) 2.2.6. Tensile tests: since tensile strength test usually gives high variation, how many duplicates of each type of sample for the tensile test should be mentioned. Statistical analysis to determine significant difference between samples in Figure 7 is recommended.
(5) 3.1. Morphological analysis and Figure 1: it is very hard to imagine how the SEM imgaes taken and what information the SEM images tell from Fig. 1. It is recommended that the authors add the regular pictures of EVA/HFB samples to see how the composite looks like and add a SEM image of pure EVA sample for comparison.
(6) The Tg transition should also be shown in Figure 2.
(7) 3.4 Electrical conductivity measurements: "The conductivity properties are quite lower compared with those reported for" should be " quite low compared with those reported for" or "relatively lower than those reported for".
(8) 3.4 Electrical conductivity measurements and Figure 5: Discussion of Fig.5 about how the pressure affect conductivity is missing and should be added. The reason and significance of showing the pressure vs conductivity should also be discussed. The data points of conductivity from the references can actually be added to Figure 5 to better show the differences.
Reviewer 2 Report
The manuscript “Ethylene-Vinyl Acetate (EVA)-Containing Waste Hemp-derived Biochar Fibers: Mechanical, Electrical, Thermal and Tribological behavior” reports synthesis and fabrication of biochar reinforced copolymer biocomposite and characterizes its effectiveness for mechanical (tensile, tribology), thermal (DSC, TGA, Conductivity, and electrical(conductivity) properties. This manuscript needs to be revisited and make corrections which might make it acceptable for publication, thus the manuscript cannot be accepted in present form. Please find my comments below.
1. The manuscript is written well, and the language is clear, however there are some typos and some sentences which are not very clear and are not conveying its intended meaning properly. Hence proofreading with a professional English speaker might help.
2. The abstract summarizes the content of paper explaining in simple words fabrication, various characterization and brief all quantitative results. The abstract needs to be rewritten.
3. In the literature survey of introduction section, the authors have not discussed biochar reinforced thermosets/thermoplastic composites, which is the focus of this paper. Authors should investigate works of the researchers mentioned below which will sustain a strong hypothesis for this work.
Effective reinforcement of engineered sustainable biochar carbon for 3D printed polypropylene biocomposites. Composites Part C: Open Access 7 (2022): 100221.
3D printing of spent coffee ground derived biochar reinforced epoxy composites. Journal of Composite Materials 55.25 (2021): 3651-3660.
4. The main drawback of this manuscript is missing hypothesis, the authors should clearly indicate what motivated them to fabricate this composite, back it with some useful properties of biochar. The authors can refer following work.
Low temperature plasma treatment of rice husk derived hybrid silica/carbon biochar using different gas sources." Materials Letters 292 (2021): 129678.
5. In the materials section, clearly indicate if the hemp fibers were provided or biochar was provided by Assocanapoa.
6. The pyrolysis method should be in methods section, if done as a part of this work to give clarity to readers.
7. Authors mention “different weight HFB loadings (namely, 5, 10, 20 and 30 wt.%) were prepared.” But authors have consistently reported results of 40 wt.% loading?
8. Please add either a schematic or real pictures of fabrication process, it will give clarity to readers. Use the literature mentioned above as a reference.
9. During explanation authors mentioned distribution of biochar, how is it possible to observe 50 µm biochar distribution in 10 and 20 µm scaled SEM image. These images are good only for matrix attachment, fiber pullout etc.
10. The authors extensively explained the DSC thermal behaviors of neat polymer which is not the focus of this work, however there is no explanation on why DSC thermal behavior did not improve, the more appropriate reason could be due to there was no covalent interaction between filler and matrix.
11. For TGA analysis the authors seem to have interesting results but did not highlight the important part, compare temperatures at 50% weight, Rate of maximum decomposition from derivative curves gives interesting observation but no mention in results.
12. For thermal conductivity the authors gave the reason for reduced thermal conductivity up to 10%, as improper interface and air pockets which might not be true, as with increasing wt.% the improper interface and air pockets will also increase, unless authors found otherwise.
13. In tensile results, Authors mention “Moreover, the increased stiffness results in lower elongation at break (fig 7B), this can’t be true, the real reason for this behavior could be either reduced crystallinity or improper interface between filler and matrix which can be observed in SEM figures too, leading to sudden brittle failure.
14. The naming of figures in figure 9. Is very confusing. Why are authors showing figures of alumina ball? The alumina ball should not be affected by friction. Please justify.
Round 2
Reviewer 1 Report
This paper has been greatly improved. This paper can be accepted after a minor revision.
Fig. 2: The place of the Tg transition looks like a straight line. The features of Tg transition cannot be seen. It is suggested the authors partially enlarge the graph at the Tg to better show the Tg transition.
Author Response
We have introduced an inset in figure 2A to better show the Tg transition of the composites